# A semi-automated cell tracking protocol for quantitative analyses of neutrophil swarming to sterile and *S. aureus* contaminated bone implants in a mouse femur model

**Sashank Lekkala**[1,2], **Youliang Ren**[2,3], **Jason Weeks**[2], **Kevin Lee**[1,2], **Allie Jia Hui Tay**[1,2], **Bei Liu**[2], **Thomas Xue**[1,2], **Joshua Rainbolt**[2], **Chao Xie**[2,3], **Edward M. Schwarz**[1,2,3]*, **Shu-Chi A. Yeh**[1,2,3,4]*

1 Department of Biomedical Engineering, University of Rochester, Rochester, New York, United States of America, 2 Center for Musculoskeletal Research, University of Rochester Medical Center, Rochester, New York, United States of America, 3 Department of Orthopaedics and Rehabilitation, University of Rochester Medical Center, Rochester, New York, United States of America, 4 Department of Physiology/Pharmacology, University of Rochester Medical Center, Rochester, New York, United States of America

* ShuChi_Yeh@URMC.Rochester.edu (SCAY); Edward_Schwarz@URMC.rochester.edu (EMS)

**Data Availability Statement:** All relevant data are within the manuscript and its Supporting information files. The MATLAB codes used for this

## Abstract

Implant-associated osteomyelitis remains a major orthopaedic problem. As neutrophil swarming to the surgical site is a critical host response to prevent infection, visualization and quantification of this dynamic behavior at the native microenvironment of infection will elucidate previously unrecognized mechanisms central to understanding the host response. We recently developed longitudinal intravital imaging of the bone marrow (LIMB) to visualize host cells and fluorescent *S. aureus* on a contaminated transfemoral implant in live mice, which allows for direct visualization of bacteria colonization of the implant and host cellular responses using two-photon laser scanning microscopy. To the end of rigorous and reproducible quantitative outcomes of neutrophil swarming kinetics in this model, we developed a protocol for robust segmentation, tracking, and quantifications of neutrophil dynamics adapted from Trainable Weka Segmentation and Track-Mate, two readily available Fiji/ImageJ plugins. In this work, *Catchup* mice with tdTomato expressing neutrophils received a transfemoral pin with or without ECFP/EGFP-expressing USA300 methicillin-resistant *Staphylococcus aureus* (MRSA) to obtain 30-minute LIMB videos at 2-, 4-, and 6-hours post-implantation. The developed semi-automated neutrophil tracking protocol was executed independently by two users to quantify the distance, displacement, speed, velocity, and directionality of the target cells. The results revealed high inter-user reliability for all outcomes (ICC > 0.96; p > 0.05). Consistent with the established paradigm on increased neutrophil swarming during active infection, the results also demonstrated increased neutrophil speed and velocity at all measured time points, and increased displacement at later time points (6 hours) in infected versus uninfected mice (p < 0.05). Neutrophils and bacteria also exhibit directionality during migration in the infected mice. The semi-automated cell tracking protocol provides a

study are available on GitHub: Lee, K., Lekkala, S., & Yeh, S. A. Quantification of tracks generated from TrackMate [Computer software]. https://github.com/SashankLekkala/Semi-automated-cell-tracking. DOI: 10.5281/zenodo.10814082.

**Funding:** Part of this work was supported by the AOTrauma Clinical Priority Program, and the National Institutes of Health: R21 AR081050 (C. Xie), P30 AR069655 (E. Schwarz) & P50 AR072000 (E. Schwarz). The funders had no role in study design, data collection and analysis, decision to publish, or preparation of the manuscript.

**Competing interests:** The authors have declared that no competing interests exist.

streamlined approach to robustly identify and track individual cells across diverse experimental settings and eliminates inter-observer variability.

## Introduction

Implant-associated osteomyelitis remains one of the most prevalent and serious orthopaedic problems: the incidences of infection for all orthopaedic subspecialties range from 0.1%-30%, at a cost of $17,000-$150,000 per patient [1]. As neutrophil swarming to the surgical site is a critical host response to prevent infection, visualization and quantifying this dynamic behavior at the native microenvironment of infection will provide previously unrecognized mechanisms central to understanding the host response that may not be fully recapitulated ex vivo [2].

Advances in intravital microscopy of long bones, such as the utilization of Gradient Index (GRIN) lenses, have facilitated deep-tissue visualization including longitudinal imaging of the bone marrow (LIMB) deep within the femur [3]. Specifically, we previously described LIMB which allows direct visualization of fluorescent host cells and bacteria proximal to a transfemoral implant in mice [4]. These imaging systems facilitate the investigation of immune mechanisms, pathogen evasion strategies, and the effects of different therapeutic agents on the kinetics of immune cells. To quantify neutrophil swarming kinetics, TrackMate [5, 6], an open-source, user-friendly plugin available within Fiji [7] is an attractive option that allows interactive cell tracking. Previous studies have reported diverse approaches to counting and tracking cells using TrackMate [8–12]. However, some of these tracking approaches utilize computationally expensive models or are restricted to specific operating systems [12]. The most apparent cellular architecture in the bone infection setting is the accumulation of bacteria and host immune cells over time. In this regard, these protocols are sub-optimal in segmenting and tracking densely populated cells with variable shapes and fluorescence intensities which became even more problematic in the context of three-dimensional intravital multiphoton laser scanning microscopy (IV-MLSM). While manual tracking options are available [6, 12], they are subject to significant inter-user variation. Subsequent statistical analyses are therefore not meaningful given such inter-user variability.

To overcome these limitations, we developed post-processing workflow, and protocols taking advantage of Trainable Weka Segmentation (TWS) [13], a machine learning tool available within Fiji, to accurately segment the cells from the background before feeding the data to TrackMate. This automatic segmentation eliminated the need for user-defined variables, such as thresholding, noise removal, etc., and largely minimized inter-user variability. Moreover, a classifier trained by the user to recognize the cells of interest can be created using TWS, which can be used to batch-process videos for high-throughput analyses. The developed protocol enabled robust cell tracking in IV-MLSM timelapse videos of neutrophils adjacent to infected and uninfected femoral implants in vivo, and their interactions with bacteria.

## Methods

### MRSA strain and implants

The most prevalent community-acquired methicillin-resistant *Staphylococcus aureus* (MRSA) strain, USA300, was used for all experiments. The bacteria were grown on tryptic soy agar (TSA) or in tryptic soy broth (TSB) at 37°C. We transformed USA300 LAC (ATCC AH1680) with the pCM29-*sarA::ecfp* or pCM29-*sarA::egfp* reporter plasmids to generate ECFP and EGFP expressing USA300 strains, respectively, as we previously reported [14]. Positive pCM29 plasmid transformation renders USA300 LAC resistant to chloramphenicol. Therefore, ECFP+

and EGFP$^+$ USA300 LAC transformants were positively selected in TSB with 10 μg/mL chloramphenicol. Subsequently, fluorescence microscopy was used to confirm the positive transformation of ECFP and EGFP into the USA300 LAC strain [4].

A flat titanium wire (cross-section 0.2 mm × 0.5 mm; MicroDyne Technologies, Plainville, CT) was cut to 4 mm length and bent into an L-shaped implant: long side 3 mm, short side 1 mm [15]. After sterilization, the implant was incubated in the overnight culture of transformed USA300 for 30 minutes prior to the implantation procedure as previously described [4].

## Animal surgery and LIMB

All animal research was performed under protocols approved by the University of Rochester Committee on Animal Resources (UCAR-2019-015). We obtained Catchup mice (C57BL/6 genetic background) [16] from Dr. Minsoo Kim (University of Rochester Medical Center) and maintained the colony. For this study, we used both male and female mice that were 12–20 weeks of age. Thirty minutes before surgery, the mice were given Buprenorphine SR (1 mg/kg) subcutaneously. The mice were then anesthetized with xylazine (10 mg/kg) and ketamine (100 mg/kg) administered intraperitoneally. The surgical procedure and the imaging setup were based on protocols previously described [4]. Briefly, the right femur was implanted with a customized LIMB system. The implant enabled imaging of a fixed region of interest (ROI) proximal to an L-shaped transcortical pin in the diaphyseal bone marrow. The mice received either a sterile pin or a USA300 contaminated pin.

The mice were imaged at 2-, 4-, and 6-hours post-implantation while anesthetized. During imaging, the mice were lying prone, and the LIMB implant was connected to a custom adaptor for proper placement of the objective lens to find the same ROI. The animals were euthanized by $CO_2$ overdose followed by cervical dislocation to ensure euthanasia.

## Intravital two-photon laser scanning microscopy

IV-MLSM was performed using an Olympus FVMPE-RS multiphoton imaging system (Olympus), equipped with MaiTai and InsightX3 Titanium:Sapphire lasers (Spectra-Physics, Santa Clara, CA), and an LUCPLFLN 20× (NA 0.45) air objective (Olympus, Tokyo, Japan). The lasers were tuned to 1050 nm and 860 nm and the fluorescence of ECFP/EGFP, and tdTomato were collected with 460–500 nm/495–540 nm, and 575–630 nm filters, respectively. Images were acquired at a size of 512 × 512 pixels with a 0.01-μs pixel dwell time. At 2, 4, and 6 hours after implantation, three-dimensional (424.26×424.26×60 μm$^3$), time-lapse (30-second interval for 30 minutes) image stacks were acquired using FLUOVIEW (31S-SW, Olympus).

## Cell tracking and image quantification

We developed a semi-automated protocol to track neutrophils and quantify their characteristics of cell motility (Fig 1A, see S1 File for a step-by-step guide). In brief, maximum intensity projection was used to project the three-dimensional image stacks in 2D given that the cells of interest and bacteria are primarily on the surface of the implant. The image stacks were drift-corrected using the Image Stabilizer plugin [17]. To remove noise and non-moving artifacts contributed by autofluorescent objects, the minimum intensity projection of the time-lapse images was subtracted from all the slices (Fig 1C). To avoid inter-user variability primarily introduced by inaccurate cell identification during the tracking process, the neutrophils were first segmented using TWS, which uses a library of machine learning training features to generate a probability score of the foreground (Fig 1D) [13]. The training features for the TWS classifier can be categorized into noise removal, edge detection, and texture description features [13]. We compared various combinations of these features focusing on noise removal

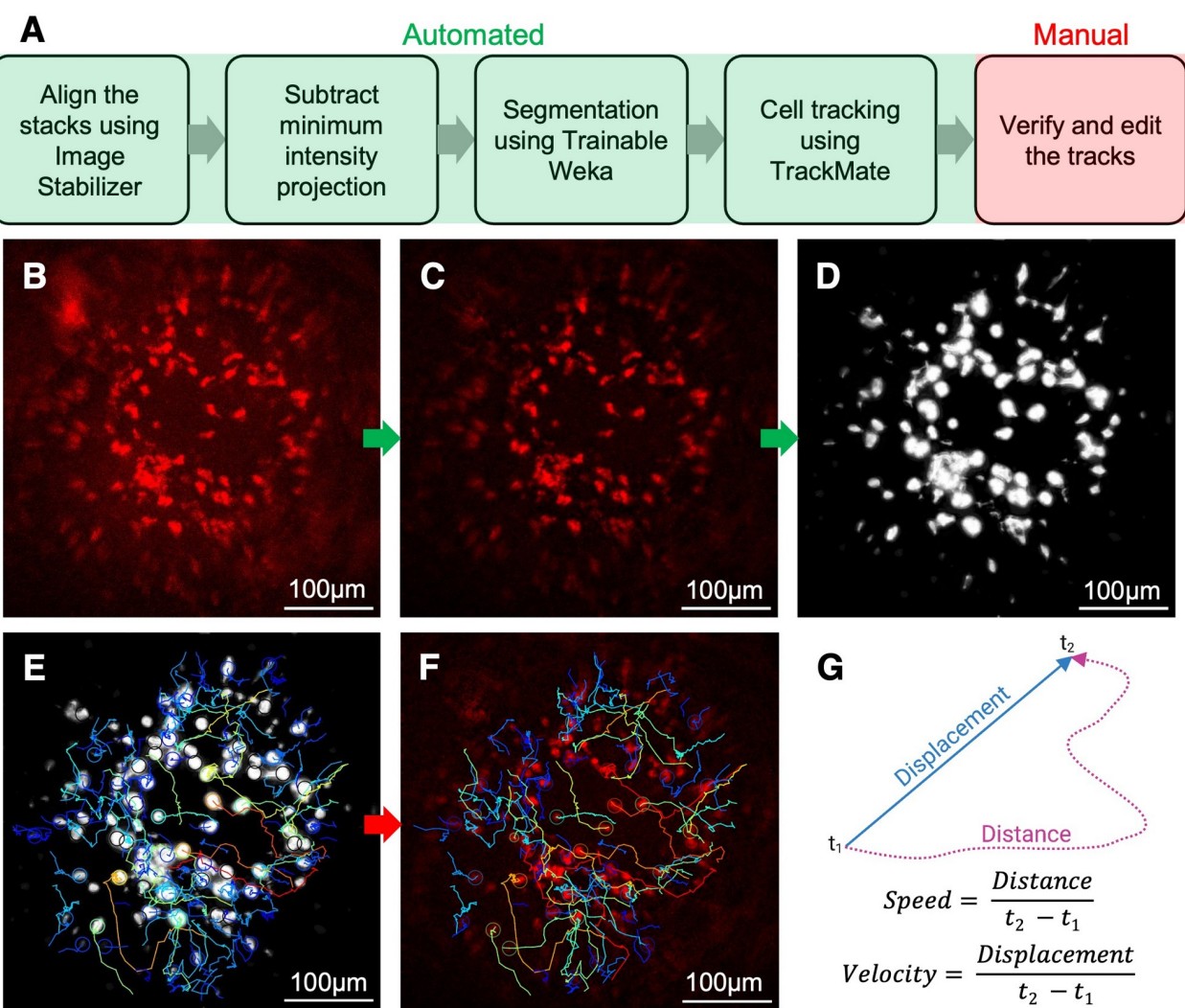

**Fig 1. Workflow for cell tracking using trainable Weka segmentation and TrackMate in ImageJ.** (A) Flowchart of the protocol. (B) Representative 2D LIMB image from a 30-minute timelapse video (Scale bar = 100μm) (C) The minimum intensity projection of the timelapse stack was subtracted from all the slices to reduce noise and remove non-moving artifacts. (D) A custom classifier trained to detect neutrophils was applied to create probability maps using Trainable Weka Segmentation (TWS). (E) In TrackMate, the cells were detected using a Laplacian of Gaussian (LoG) detector, and the tracks were determined by a linear assignment problem-based (LAP) tracker. (F) These tracks were overlaid on (C) and edited manually for accuracy. (G) A custom MATLAB code was used to calculate the distance and displacement for each neutrophil track. Speed and velocity were calculated as the ratios of distance and displacement to track duration, respectively. Directionality was calculated as the ratio of displacement and distance. Representative images from N = 6 mice.

and edge detection to optimize cell detection and tracking (S2 Appendix in S1 File). The best-performing classifier incorporated Gaussian blur, Sobel filter, Hessian, and Difference of Gaussians training features with a minimum sigma of 1 and a maximum sigma of 16. For classification, we used Fast Random Forest, the default machine learning model in TWS. After identifying the best training features, we refined the labels for background and neutrophils to arrive at an optimal classifier which was consistently employed for all timelapse videos. The probability maps generated were then used as input for TrackMate to track moving cells.

Specifically, the moving objects are defined in TrackMate based on cell diameter and total displacement. The cells were detected using a Laplacian of Gaussian (LoG) detector, and the

tracks were determined by a linear assignment problem-based (LAP) tracker (Fig 1E) [5, 6]. The maximum linking distance was set to 20μm based on ground truth provided by manual tracking. Note that this distance is determined empirically by the time interval of longitudinal imaging (30 seconds in our case). After the tracks were computed, a displacement filter of 16μm was set to remove tracks that are a result of minor intensity fluctuations in stationary cells. These tracks were overlaid on the drift-corrected stack and edited manually as needed for accuracy (Fig 1F). A custom MATLAB code (S3 Appendix in S1 File) was used to calculate the distance and displacement for each neutrophil track. Speed and velocity were calculated as the ratios of distance and displacement to track duration, respectively (Fig 1G). Directionality, a measure of the straightness of a track was calculated as the ratio of displacement and distance.

Using this protocol, we also tracked bacteria by employing a different TWS classifier and changing the particle size to 8μm. To study the migration pattern and interactions of neutrophils and bacteria, a custom MATLAB script was developed to map and track spatial associations of neutrophils and bacteria based on their migration trajectories (S4 Appendix in S1 File). The script also identifies cell interaction events based on criteria that the neutrophils and bacteria were in proximity (within 16μm distance) for a prolonged duration ($\geq$ 5 time frames, 150 seconds).

The volume of neutrophils proximal to the implant was quantified using Imaris (Oxford Instruments). Briefly, the 3D stack was smoothed, the background was subtracted, and a user-defined threshold was applied to segment neutrophils. Then the imaging artifacts were manually removed, and the neutrophil volume was calculated based on the number of voxels (S1 Fig in S1 File).

## Statistics

Data are presented as median ± interquartile range. Mann-Whitney tests adjusted for multiple comparisons by the Holm-Šídák method were used to test if the tracking metrics differed between individuals and the study groups. Interclass correlation coefficients (ICC) were calculated based on two-way random effects model with absolute agreement for two raters [18].

## Results

### Reproducibility of the tracking protocol

The major limitation of manual tracking is high inter-user variability. The two users assigned to analyze the same IV-MLSM video often identified different tracks. The lack of consistency is shown in the data distribution, where the median displacement and velocity of neutrophils differed by 25% and 22% respectively between the two users (S2 Fig in S1 File). While these results were not statistically significant, such substantial numerical differences impede reliable conclusions. Furthermore, the inter-class correlation coefficient for manual tracking was as low as 0.85, indicating only a modest user agreement.

In contrast, our protocol yields reproducible tracking metrics from two users who independently analyzed three IV-MLSM videos. While there were minor differences in the tracks generated by both users (Fig 2, S3 Fig in S1 File), we obtained robust inter-reader reliability (ICC > 0.96). The track metrics and data distribution are in good agreement (p > 0.05) (Fig 2C–2G, S3C-S3G Fig in S1 File).

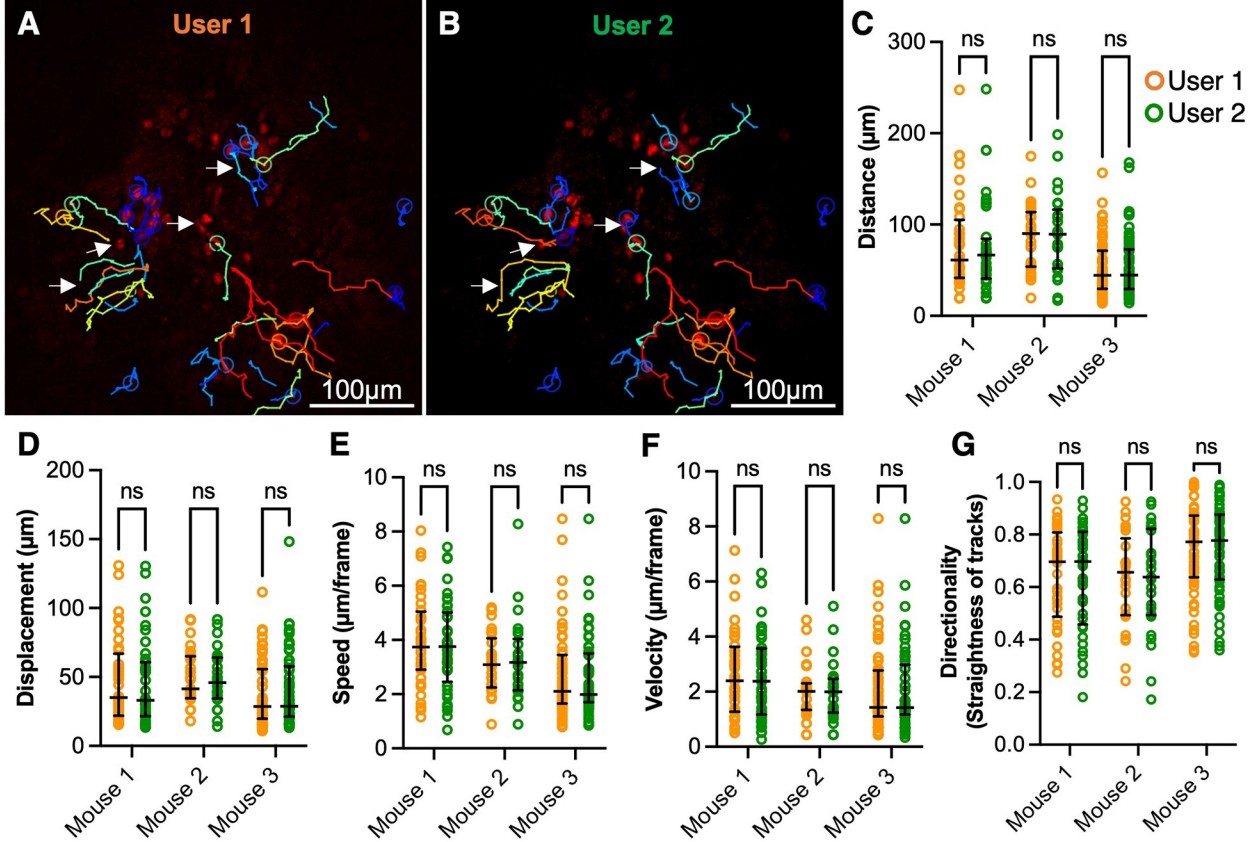

**Fig 2. The tracking protocol resulted in low inter-person variability in generating tracks.** Two users independently analyzed three IV-MLSM timelapse videos and the generated track parameters were compared. Representative 2D IV-MLSM image with overlaid tracks generated by user 1 (A) and user 2 (B). The differences in the tracks generated by both users are shown with white arrows. Semiautomated quantification of neutrophil distance traveled (C), displacement (D), mean speed (E), mean velocity (F), and directionality (G), was performed and the data are presented with the median and interquartile range. (ns = not significant as determined by Mann-Whitney tests adjusted by the Holm-Šídák method (n = 25–57 tracks. Representative images from N = 3 mice)).

## Quantification of neutrophil swarming proximal to infected vs. sterile implant

As it has been well established that neutrophils demonstrate increased swarming in the setting of an active infection [2, 19, 20], we validated our protocols by comparing neutrophil behavior proximal to MRSA infected versus sterile femoral implants. Consistent with the infection status, the neutrophil volume proximal to the pin increased with time in the infected animals but remained consistent in uninfected animals (Fig 3A and 3B, S1–S3 Videos). However, these results were not statistically significant. The neutrophils in the infected animals traveled longer distances at 6 hours and longer displacements at 4 hours and 6 hours compared to the neutrophils in uninfected animals (Fig 3C and 3D). Despite the path length of migration increasing several hours after implantation, the neutrophils in the infected animals traveled at significantly greater speeds (distance/time) and velocities (displacement/time) at all time points compared to uninfected animals (Fig 3E and 3F). In agreement, the directionality of the neutrophil migration in the infected animals, indicative of swarming behaviors, was shown at the early time point (2 hours) compared to uninfected animals (Fig 3G).

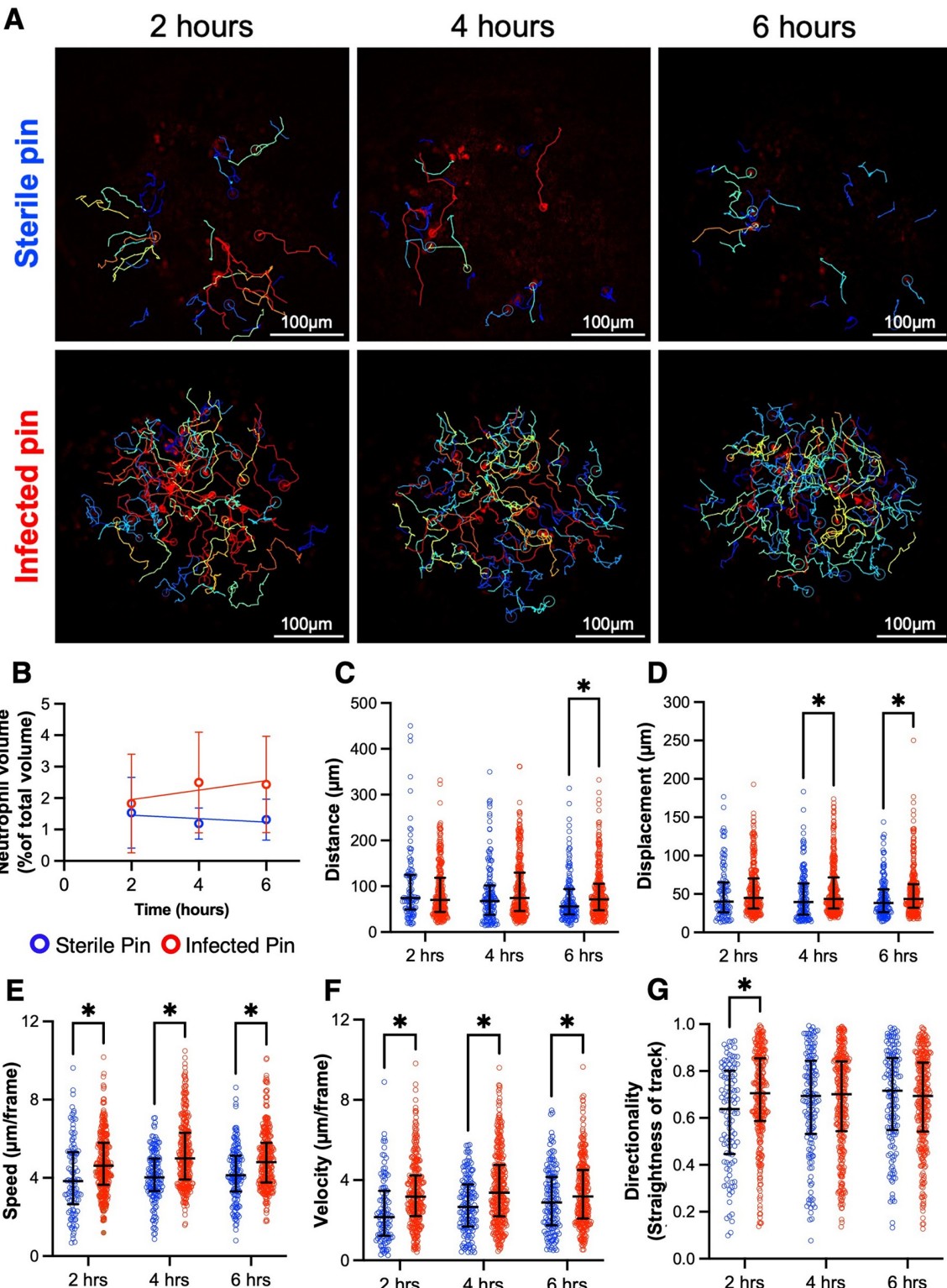

**Fig 3. Increased neutrophil swarming proximal to MRSA contaminated vs. sterile bone implant.** *Catchup* mice were challenged with a sterile or MRSA-contaminated transfemoral implant and neutrophil swarming behaviors were quantified from 30min IV-MLSM videos obtained at the indicated time post-implantation. (A) Representative 2D IV-MLSM images with overlaid neutrophil tracks proximal to sterile and infected pin at 2-, 4-, and 6-hours post-implantation (S1–S3 Videos). (B) Change in neutrophil volume with time proximal to infected and sterile implants calculated using Imaris (S1 Fig in S1 File). Data is shown as mean ± standard deviation.

Unpaired t-tests were used to test the differences between infected and sterile conditions (n = 3). Semiautomated quantification of neutrophil distance traveled (C), displacement (D), mean speed (E), mean velocity (F), and directionality (G), was performed and the data are presented with the median and interquartile range. (*p < 0.05 as determined by Mann-Whitney tests adjusted by the Holm-Šídák method (n = 105–316 tracks/group/timepoint, N = 3 mice/group)).

Moreover, by tracking both bacteria and neutrophils, parallel migration pattern was observed (Fig 4A), where bacteria exhibited stable contact with a neutrophil, or made intermittent contact with multiple cells (Fig 4B). The contact time and frequency may be a potential metric at the early time point to correlate with phagocytosis events at later time points (colocalization of neutrophils and bacteria) and the efficacy of different therapeutic regimens on neutrophil-pathogen interactions.

## Discussion

In this paper, we described a protocol to track cells from IV-MLSM videos accurately and reproducibly. The protocol included post-processing steps to minimize stationary and motion artifacts and employed TWS, a machine learning algorithm for robust segmentation of the target cell population. The use of these segmented images greatly reduced the inter-user variability when generating tracks in TrackMate and customized Matlab code. Using these quantitative metrics, we showed that neutrophil swarming is more pronounced proximal to infected implants compared to uninfected implants.

This protocol is semi-automated with minimal user interference which resulted in high reproducibility (Fig 2). By standardizing the pre-processing and segmentation methods, the need for selecting an arbitrary threshold, a main source of inconsistencies, was eliminated. Further, utilizing a singular global threshold might not yield optimal results for neutrophil segmentation due to the inherent variability and low signal to noise ratios in fluorescence associated with 3D intravital imaging. TWS overcomes this limitation by using a collection of machine learning algorithms to produce pixel-based segmentation. In addition to automating the process, we have outlined a set of guidelines for manual track verification (S1 File) which further reduces inter-user variability.

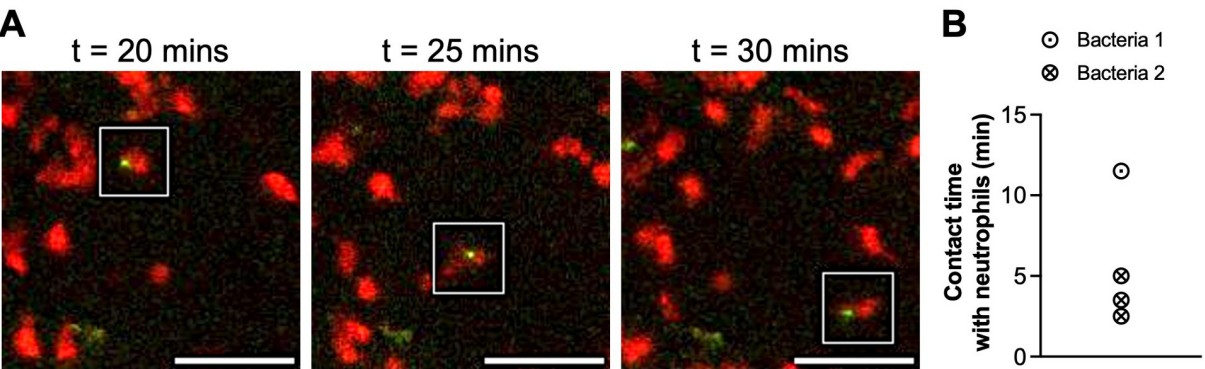

**Fig 4. Parallel migration pattern of neutrophils and bacteria.** A custom script quantified the interactions of neutrophil and bacteria tracks. (A) Representative 2D IV-MLSM images showing parallel migration of neutrophils and bacteria over several time points where the cells made stable contact with the neutrophil (Scale bar = 50μm). (B) Contact time between neutrophils and bacteria revealed stable interaction or intermittent contact with several neutrophils (S4 and S5 Videos).

We opted not to extensively denoise the image prior to segmentation to retain as many morphological and textural features as possible. Such information is a critical "training set" to identify the background in the machine learning algorithms. We subtracted the minimum intensity projection from the stack to remove any stationary artifacts. However, this method will also remove any stationary cells and the nidus. Given our focus on tracking only the moving cells, this pre-processing method proved most effective for our purpose. Adding extensive denoising steps increases the complexity of the protocol and may introduce user variability and/or denoising artifacts. This protocol can be adapted with alternative denoising techniques such as morphological filtering (for example, white top-hat) to preserve the stationary details [20]. One of the key advantages of TWS is the ability to train the system to detect the foreground even from a noisy background where the features of the background were also extracted for training. For this reason, it was shown to be the most efficient when images are directly segmented using TWS without any pre-processing.

Note that we chose a conservative classifier in TWS for its capability to detect even faintly fluorescent neutrophils. Consequently, the resulting probability maps depicted larger neutrophils compared to the original stack (Fig 1C and 1D). The approach is sufficient in detecting early swarming behaviors after the onset of infection. A potential drawback of this classifier is that when the cells are densely packed, potentially at even later time points, multiple neutrophils may appear as a single object. A more aggressive classifier may perform better in distinguishing individual cells although a trade-off of missing dim subjects is expected (S4 Fig in S1 File). Deep learning programs such as StarDist which also use object boundaries for segmentation may overcome this limitation with a custom training set [10, 21]. However, StarDist is most effective for cases of nuclear staining or cells with star-convex shapes which may not always apply to varying shapes of patrolling cells such as neutrophils [21].

The probability maps were used to generate tracks in TrackMate. For object detection, a Laplacian of Gaussian (LoG) detector which detects local maxima was used [22]. The object size was set to 16μm, slightly larger than typical sizes of neutrophils ($< 8$–15 μm). This decision is based on size fitting on several probability maps to ensure separate detection of two closely spaced neutrophils. A quality filter ($= 0.001$) which is a measure of the value of the local maxima and the size of the object was applied to remove objects detected from noise. For linking these objects, we employed a linear assignment problem-based algorithm (LAP), which links objects based on the lowest square distance [23]. This approach results in an expected limitation when two cells come close to each other, the algorithm may favor linking these two cells at the frame of contact instead of tracking the same cell. These discrepancies were manually corrected.

For the analysis, cells with low fluorescence intensity and/or low displacement are excluded. This exclusion is not a limitation but rather a conscious decision for our application. We excluded dimly fluorescent cells to exclude autofluorescent cells that are not neutrophils. These low-intensity cells can be easily included in the analysis by changing the TWS classifier to be more conservative. Similarly, we intentionally excluded cells with smaller displacement ($<16$μm) which can be included by changing the displacement filter in TrackMate.

We performed a comparative analysis between the results from manual and semi-automated tracking protocols (S5 Fig in S1 File, S6 Video). We performed manual tracking using TrackMate after performing minimum intensity projection subtraction where a user tracked individual cells (S5A Fig in S1 File). Then we compared the results with those from the semi-automated tracking protocol (S5B Fig in S1 File). Most of the tracks were similar between both cases (S5 Fig and S1 Table in S1 File). However, as expected, manual tracking resulted in tracks from faintly fluorescent cells which are potentially auto-fluorescent cells that are not neutrophils (also a source of user variability) (S6 Video). In addition, we showed in S2 Fig in S1 File

that manual tracking suffers from inter-person variability. This limitation is overcome by the semi-automated protocol, as TWS segments the cells of interest.

Cell tracking protocols perform less optimally when the density of cells is high [24]. Detection of individual neutrophils was compromised when the cells are crowded (S7 Video, Inset 1). This distinction is not possible even manually and would require other tools such as cell membrane labeling to distinguish crowded cells separately. In these cases, the cells that merged as a cluster may be misidentified as one cell, which needs to be edited manually. For such cases, we opted to end the tracks when the cells cluster, and then restart tracking after the cells re-emerged from the cluster. This edit shortens the tracks resulting in lower distance and displacement for these cells. In contrast, time-normalized metrics such as speed and velocity are less susceptible to this limitation. Additionally, it is important to note that shorter tracks tend to be straighter, which potentially biases the distribution of the directionality parameter. In this studyI, user verifications of error-prone regions (e.g., cells with small displacement, low intensity, or located at the boundaries of the field of view) are necessary. Each track took about a minute for the users to verify (across 60 frames). Previous studies proposed solutions to overcome this limitation by introducing specific rules for linking tracks when two cells collide or come close to each other [8]. However, these algorithms are computationally expensive, and these rules are specific to the cell type and imaging settings. Future optimization focusing on connection rules and penalties will help address this issue.

Several alternatives exist for segmentation and tracking. However, the key advantage of TWS and TrackMate is their ready availability as ImageJ plugins and the user-friendly interface. TWS has been successfully used for a wide range of applications and can be an attractive tool to analyze histology sections in the future [25, 26]. Machine and deep learning segmentation techniques such as StarDist, Cellpose, Ilastik, etc. are available, but they require a robust training set and classifier that needs to be generated outside of ImageJ [27, 28]. The trained classifier can be called into ImageJ to segment other similar images while additional integration of the workflow will be needed. These programs may offer better integration with Python or other coding platforms if ImageJ is not required for further analysis.

Using the LIMB system, we studied neutrophil behavior in response to sterile or MRSA-contaminated implants. As expected, the neutrophil volume proximal to the infected pin was higher compared to that of the sterile pin indicating an active response to infection. Furthermore, in the infected mice, the neutrophils traveled farther (higher distance and displacement) and faster (higher speed and velocity) especially at later time points potentially indicating more potent activation and chemotactic signals compared to the uninfected mice. Future studies with a focus on the quantification of these signals will help elucidate the mechanisms of neutrophil activation and migration. Then the tracking metrics can be validated against these biological outcomes.

Several studies performed longitudinal imaging of neutrophils and macrophages and reported chemotaxis followed by phagocytosis but did not fully quantify this behavior [29, 30]. In this study, we proposed a quantitative method to study this parallel migration pattern as an indicator of chemotactic behaviors. Using cell tracking data, we demonstrated that we could determine the number of contacting cell pairs and their contact time. Our future goal is to implement this algorithm, especially in the context of different antibiotic treatment regimens to understand the correlation between parallel migration, phagocytosis, and the therapeutic outcome. Further, morphological alteration has been shown to provide functional information on neutrophils [31], and would be an interesting aspect to interrogate. Although the morphology of the identified cells can be studied using ImageJ tools such as "Analyze particles" or "MorphoLibJ" [32], the GRIN lens-based system with a small numerical aperture does not provide sufficient image resolution to visualize cell processes and fine morphological changes.

Altogether, we developed a semi-automated tracking protocol with minimum user interference by leveraging the capabilities of TWS and TrackMate, two readily available Fiji plugins. We showed that the protocol offers low inter-user variability along with high accuracy. The protocol can be easily adapted to study the kinematics of different cells by changing the TWS classifier and the tracking parameters. The tracking protocol offers a streamlined approach to tracking individual cells accurately and efficiently in diverse experimental settings.

## Supporting information

**S1 Video. Representative IV-MLSM timelapse of the neutrophils with the tracks overlaid proximal to a sterile pin (left) and an infected pin (right) at 2 hours.** Scale bar = 100μm. Note that the undetected neutrophils had displacement lower than 16μm.
(AVI)

**S2 Video. Representative IV-MLSM timelapse of the neutrophils with the tracks overlaid proximal to a sterile pin (left) and an infected pin (right) at 4 hours.** Scale bar = 100μm. Note that the undetected neutrophils had displacement lower than 16μm.
(AVI)

**S3 Video. Representative IV-MLSM timelapse of the neutrophils with the tracks overlaid proximal to a sterile pin (left) and an infected pin (right) at 6 hours.** Scale bar = 100μm. Note that the undetected neutrophils had displacement lower than 16μm.
(AVI)

**S4 Video. Representative IV-MLSM timelapse of the neutrophils (left) and USA300 MRSA (right) with the tracks overlaid proximal to an infected pin at 2 hours.** Scale bar = 100μm. Note that the undetected neutrophils and bacteria had displacement lower than 16μm.
(AVI)

**S5 Video. Representative IV-MLSM timelapse of the neutrophils and bacteria proximal to an infected pin at 2 hours (same experiment as S1 and S2 Videos).** The insets are tracking two different bacteria. Scale bar = 100μm. Note that the undetected neutrophils and bacteria had displacement lower than 16μm.
(AVI)

**S6 Video. Comparison between manual and semi-automated tracking.** IV-MLSM timelapse of the neutrophils with the tracks overlaid for manual tracking (left) and semi-automated tracking (right). Scale bar = 100μm. Note that the manual tracking resulted in tracks from faintly fluorescent cells which is a source of user-variability.
(AVI)

**S7 Video. Representative IV-MLSM timelapse of the neutrophils with the tracks overlaid.** Note that the neutrophils in Inset 1 appear distinctly and then cluster until which time the cells cannot be tracked. Once the cells emerged from the cluster, new tracks were started for these cells. This limitation is inherent in our and previous tracking protocols. The neutrophils in the Inset 2 were excluded because they were out of focus and had lower displacement (<16μm). These cells can be included for tracking by adjusting the TWS and TrackMate parameters. Scale bar = 100μm.
(AVI)

**S1 File.**
(DOCX)

## Acknowledgments

Schematic images were generated with BioRender. The authors thank Kaye Thomas, Emma Norris, Julie Zhang, Yurong Gao, and Rhonda Jean Kay for their assistance with the LIMB imaging and analyses.

## Author Contributions

**Conceptualization:** Chao Xie, Edward M. Schwarz, Shu-Chi A. Yeh.

**Data curation:** Sashank Lekkala, Youliang Ren, Kevin Lee, Chao Xie, Edward M. Schwarz, Shu-Chi A. Yeh.

**Formal analysis:** Sashank Lekkala, Youliang Ren, Jason Weeks, Kevin Lee, Allie Jia Hui Tay, Bei Liu, Thomas Xue.

**Funding acquisition:** Chao Xie, Edward M. Schwarz, Shu-Chi A. Yeh.

**Investigation:** Chao Xie, Edward M. Schwarz, Shu-Chi A. Yeh.

**Methodology:** Sashank Lekkala, Youliang Ren, Jason Weeks, Kevin Lee, Thomas Xue, Joshua Rainbolt, Chao Xie, Edward M. Schwarz, Shu-Chi A. Yeh.

**Project administration:** Chao Xie, Edward M. Schwarz, Shu-Chi A. Yeh.

**Resources:** Chao Xie, Edward M. Schwarz, Shu-Chi A. Yeh.

**Software:** Sashank Lekkala, Jason Weeks, Kevin Lee, Shu-Chi A. Yeh.

**Supervision:** Chao Xie, Edward M. Schwarz, Shu-Chi A. Yeh.

**Validation:** Sashank Lekkala, Kevin Lee, Chao Xie, Edward M. Schwarz, Shu-Chi A. Yeh.

**Visualization:** Sashank Lekkala.

**Writing – original draft:** Sashank Lekkala.

**Writing – review & editing:** Sashank Lekkala, Youliang Ren, Jason Weeks, Kevin Lee, Allie Jia Hui Tay, Bei Liu, Thomas Xue, Joshua Rainbolt, Chao Xie, Edward M. Schwarz, Shu-Chi A. Yeh.

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
