## [Decision Letter · Decision Letter 0]

8 Jan 2024

PONE-D-23-40619A semi-automated cell tracking protocol for quantitative analyses of neutrophil swarming to sterile and S. aureus contaminated bone implants in a mouse femur modelPLOS ONE

Dear Dr. Lekkala,

Thank you for submitting your manuscript to PLOS ONE. After careful consideration, we feel that it has merit but does not fully meet PLOS ONE’s publication criteria as it currently stands. Therefore, we invite you to submit a revised version of the manuscript that addresses the points raised during the review process.

We look forward to receiving your revised manuscript.

Kind regards,

Shigao Huang

Academic Editor

PLOS ONE

3. To comply with PLOS ONE submissions requirements, in your Methods section, please provide additional information regarding the experiments involving animals and ensure you have included details on (1) methods of sacrifice, (2) methods of anesthesia and/or analgesia, and (3) efforts to alleviate suffering.

 [Part of this work was supported by the AOTrauma Clinical Priority Program, and the National Institutes of Health: R21 AR081050 (C. Xie), P30 AR069655 (E. Schwarz) & P50 AR072000 (E.Schwarz).].  

6. Please include the reference section of your manuscript.

7. We note that Figure(s), 1A, B, C, D,E, F, 2A,B, 3A, 4A, S1A,B, S2A and B in your submission contain copyrighted images. All PLOS content is published under the Creative Commons Attribution License (CC BY 4.0), which means that the manuscript, images, and Supporting Information files will be freely available online, and any third party is permitted to access, download, copy, distribute, and use these materials in any way, even commercially, with proper attribution. For more information, see our copyright guidelines: http://journals.plos.org/plosone/s/licenses-and-copyright.

a. You may seek permission from the original copyright holder of Figure(s), 1A, B, C, D,E, F, 2A,B, 3A, 4A, S1A,B, S2A and B to publish the content specifically under the CC BY 4.0 license. 

Reviewers' comments:

Reviewer's Responses to Questions

**Comments to the Author**

1. Is the manuscript technically sound, and do the data support the conclusions?

Reviewer #1: Yes

Reviewer #2: Yes

2. Has the statistical analysis been performed appropriately and rigorously? 

Reviewer #1: Yes

Reviewer #2: Yes

3. Have the authors made all data underlying the findings in their manuscript fully available?

Reviewer #1: Yes

Reviewer #2: Yes

4. Is the manuscript presented in an intelligible fashion and written in standard English?

Reviewer #1: Yes

Reviewer #2: Yes

5. Review Comments to the Author

Reviewer #1: The manuscript titled "A Semi-Automated Cell Tracking Protocol for Quantitative Analyses of Neutrophil Swarming to Sterile and S. aureus Contaminated Bone Implants in a Mouse Femur Model" presents a well-structured protocol for accurate and reproducible cell tracking in IV-MLSM videos. The semiautomated tracking protocol utilizing trainable Weka segmentation (TWS) and TrackMate effectively minimizes user intervention and offers an efficient method for tracking individual cells in diverse experimental scenarios. Nevertheless, to further enhance the manuscript's quality, I recommend the authors to address certain key points:

1. In the Results section, the "Parallel Migration Pattern of Neutrophils and Bacteria" is intriguing. I think a more in-depth analysis of cell interactions would significantly enrich the understanding of this phenomenon.

2. The Discussion section mentions that "some of the cells were not tracked by the protocol." It would be beneficial for the authors to explore and clarify the reasons behind this, assessing whether it introduces any bias in the interpretation of the results.

3. To bolster the reliability of the protocol, a comparative analysis is advised. Comparing the results of manual analysis performed by a group of users with those obtained from the same IV-MLSM video using the semiautomated protocol would provide valuable insights.

4. In the Methods section, under "MRSA Strain and Implants," a comprehensive description of the process and conditions is needed.

5. In the "Animal Surgery and LIMB" subsection, it is crucial to specify the mouse strain used and detail the rearing conditions.

6. Regarding Figure 2, does "n=25-57 tracks" introduce potential statistical bias? An elaboration on this aspect would be beneficial.

7. The legend for Figure 2 should include detailed explanations of terms such as "ns" to improve reader comprehension.

8. The statistical results in Figure 3B need a more thorough description to enhance clarity and understanding.

9. On line 279, "error-prone regions (e.g., cells with small displacement, low intensity, or located at the boundaries of the field of view)" are mentioned. Given these limitations, it is crucial to discuss how the reliability of the results is assured through user verification and the protocol's capabilities.

10. While the protocol focuses on tracking, it would be interesting to see if it can be extended to analyze cell behavior, such as changes in cell morphology or interactions with other cells.

11. The manuscript mentions not extensively denoising images before segmentation. However, a more detailed rationale behind this choice and its impact on the results would add clarity.

12. What are the advantages of using machine analysis methods such as TWS and TrackMate for cell detection and tracking compared to conventional methods? Furthermore, what is the significance of the newly developed machine computation methods? A detailed discussion on these aspects is recommended.

Reviewer #2: This manuscript presents a novel protocol for retinal intravital microscopy, enabling direct observation of the effects of fluorescent Staphylococcus aureus on contaminated transfemoral implants and host cells in mice. By utilizing two-photon laser scanning microscopy, this approach allows direct visualization of bacterial colonization of implants and host cell responses. To quantify this process, the authors developed a semiautomated machine learning model to segment, track, and analyze the dynamics of neutrophil cells, quantifying their motility characteristics. The use of the trainable Weka Segmentation (TWS) machine learning tool for detecting and tracking individual cells among various cell types represents an improvement over previous cell tracking methods, particularly in segmenting and tracking cells of diverse shapes and fluorescence intensities. This advancement resulted in less variability and greater statistical significance in subsequent analyses. While the manuscript is concise, logical, and well-structured, there are areas that require further elaboration and improvement:

A detailed explanation of the origin and the basis for the application analysis of the TWS machine learning tool developed in the study is needed.

The manuscript should specify the activity state, detection area, and orientation of the mice during fluorescence detection using the two-photon laser scanning microscope.

The article mentions that individual differences can impact research results, yet the manuscript lacks detailed information on the conditions of the mice. This should be addressed.

The possibility of excessive clustering in the study of cell and bacteria interactions raises the question of how individual neutrophils are located and detected. This aspect needs clarification.

For Figure G, it is advised to maintain font size consistency with the rest of the figures.

Technical Specifications and Settings: More detailed information on the technical specifications and settings used during the imaging and analysis process would be beneficial. This should include details on the parameters of the machine learning algorithms, image acquisition settings, and any preprocessing steps taken.

Reproducibility in Different Experimental Settings: The manuscript would benefit from additional information on the protocol’s reproducibility under various experimental conditions, such as different types of tissues or varying levels of inflammation.

Future applications of the TWS machine learning tool should be outlined, providing a reasoned projection of its potential use in other directions.

6. PLOS authors have the option to publish the peer review history of their article (what does this mean?). If published, this will include your full peer review and any attached files.

Reviewer #1: **Yes: **yicheng zhao

Reviewer #2: No

---

## [Author Response · Author response to Decision Letter 0]

19 Mar 2024

We are most grateful to the Reviewers for their helpful comments, which we address below. The Reviewers’ queries are numbered and repeated in black, bold, and italicized text, and our responses follow in blue text. New text in the manuscript is indicated in blue text.

Reviewer #1: The manuscript titled "A Semi-Automated Cell Tracking Protocol for Quantitative Analyses of Neutrophil Swarming to Sterile and S. aureus Contaminated Bone Implants in a Mouse Femur Model" presents a well-structured protocol for accurate and reproducible cell tracking in IV-MLSM videos. The semiautomated tracking protocol utilizing trainable Weka segmentation (TWS) and TrackMate effectively minimizes user intervention and offers an efficient method for tracking individual cells in diverse experimental scenarios. Nevertheless, to further enhance the manuscript's quality, I recommend the authors to address certain key points:

1. In the Results section, the "Parallel Migration Pattern of Neutrophils and Bacteria" is intriguing. I think a more in-depth analysis of cell interactions would significantly enrich the understanding of this phenomenon.

We thank the Reviewer for the praise of our work and suggestions to improve the manuscript. We are currently planning dedicated biological studies that may manipulate chemotactic behaviors to further analyze this phenomenon. This part of the experiment however requires further optimization on the biology side (e.g. imaging time point, treatment dosage, etc) and is beyond the scope of this manuscript that focuses on the development of image analysis protocols. We appreciate that the Reviewer agreed on the value of parallel migration pattern, thus we have added the following paragraph to the Discussion section to elaborate on the potential applications (pages 17, lines 360-367):

“Several studies performed longitudinal imaging of neutrophils and macrophages and reported chemotaxis followed by phagocytosis but did not fully quantify this behavior.(1,2) In this study, we proposed a quantitative method to study this parallel migration pattern as an indicator of chemotactic behaviors. Using cell tracking data, we demonstrated that we could determine the number of contacting cell pairs and their contact time. Our future goal is to implement this algorithm, especially in the context of different antibiotic treatment regimens to understand the correlation between parallel migration, phagocytosis, and the therapeutic outcome.”

2. The Discussion section mentions that "some of the cells were not tracked by the protocol." It would be beneficial for the authors to explore and clarify the reasons behind this, assessing whether it introduces any bias in the interpretation of the results.

We thank the Reviewer for pointing this out, and we have clarified this further in the Discussion section (pages 14-15, lines 303-308 and 319-326), regarding the two scenarios when the cells are not tracked and the associated impact on data interpretation. 

“For the analysis, cells with low fluorescence intensity and/or low displacement are excluded. This exclusion is not a limitation but rather a conscious decision for our application. We excluded dimly fluorescent cells to exclude autofluorescent cells that are not neutrophils. These low-intensity cells can be easily included in the analysis by changing the TWS classifier to be more conservative. Similarly, we intentionally excluded cells with smaller displacement (<16µm) which can be included by changing the displacement filter in TrackMate.” 

“However, the limitation of the protocol is that when the cells are crowded, the neutrophils cannot be distinguished from one another (S7 Video, Inset 1). This distinction is not possible even manually and would require other tools such as cell membrane labeling to distinguish crowded cells separately. In these cases, the cells that merged as a cluster may be misidentified as one cell, which needs to be edited manually. For such cases, we opted to end the tracks when the cells cluster, and then restart tracking after the cells re-emerged from the cluster. This edit shortens the tracks resulting in lower distance and displacement for these cells. In contrast, time-normalized metrics such as speed and velocity are less susceptible to this limitation.” 

S7 Video. Representative IV-MLSM timelapse of the neutrophils with the tracks overlaid. Note that the neutrophils in Inset 1 appear distinctly and then cluster until which time the cells cannot be tracked. Once the cells emerged from the cluster, new tracks were started for these cells. This limitation is inherent in our and previous tracking protocols. The neutrophils in the Inset 2 were excluded because they were out of focus and had lower displacement (<16µm). These cells can be included for tracking by adjusting the TWS and TrackMate parameters. Scale bar = 100µm.

3. To bolster the reliability of the protocol, a comparative analysis is advised. Comparing the results of manual analysis performed by a group of users with those obtained from the same IV-MLSM video using the semiautomated protocol would provide valuable insights.

As suggested by the Reviewer, we performed a comparative analysis between manual and semi-automated tracking and discussed the results in the Discussion section. (pages 14-15, lines 309-318)

“We performed a comparative analysis between the results from manual and semi-automated tracking protocols (S5 Fig, S6 Video). We performed manual tracking using TrackMate after performing minimum intensity projection subtraction where a user tracked individual cells (S5 Fig A). Then we compared the results with those from the semi-automated tracking protocol (S5 Fig B). Most of the tracks were similar between both cases (S5 Fig, S1 Table). However, as expected, manual tracking resulted in tracks from faintly fluorescent cells which are potentially auto-fluorescent cells that are not neutrophils (also a source of user variability) (S6 Video). In addition, we showed in S2 Fig that manual tracking suffers from inter-person variability. This limitation is overcome by the semi-automated protocol, as TWS segments the cells of interest.”

S5 Fig. Comparison between manual and semi-automated tracking protocols. An IV-MLSM timelapse video was analyzed by manual tracking after subtracting the minimum intensity projection from the stack. The same video was analyzed using the semi-automated tracking protocol. A representative 2D IV-MLSM image with overlaid tracks generated by manual tracking is shown in (A) and semi-automated tracking is shown in (B). Quantification of neutrophil distance traveled (C), displacement (D), mean speed (E), mean velocity (F), and directionality (G), was performed, and the data are presented with the median and interquartile range. (ns = not significant as determined by Mann-Whitney tests (n = 24 tracks for manual tracking and 22 tracks for semi-automated tracking)). 

S6 Video. Comparison between manual and semi-automated tracking. IV-MLSM timelapse of the neutrophils with the tracks overlaid for manual tracking (left) and semi-automated tracking (right). Scale bar = 100µm. Note that the manual tracking resulted in tracks from faintly fluorescent cells which is a source of user-variability.

4. In the Methods section, under "MRSA Strain and Implants," a comprehensive description of the process and conditions is needed.

We thank the Reviewer for pointing this out. As suggested, we have added additional details to the methods (page 5, lines 93-106):

“The most prevalent community-acquired methicillin-resistant Staphylococcus aureus (MRSA) strain, USA300, was used for all experiments. The bacteria were grown on tryptic soy agar (TSA) or in tryptic soy broth (TSB) at 37°C. We transformed USA300 LAC (ATCC AH1680) with the pCM29-sarA::ecfp or pCM29-sarA::egfp reporter plasmids to generate ECFP and EGFP expressing USA300 strains, respectively, as we previously reported.(3) Positive pCM29 plasmid transformation renders USA300 LAC resistant to chloramphenicol. Therefore, ECFP+ and EGFP+ USA300 LAC transformants were positively selected in TSB with 10 μg/mL chloramphenicol. Subsequently, fluorescence microscopy was used to confirm the positive transformation of ECFP and EGFP into the USA300 LAC strain.(4) 

A flat titanium wire (cross-section 0.2 mm × 0.5 mm; MicroDyne Technologies, Plainville, CT) was cut to 4 mm length and bent into an L-shaped implant: long side 3 mm, short side 1 mm.(5) After sterilization, the implant was incubated in the overnight culture of transformed USA300 for 30 minutes prior to the implantation procedure as previously described.(4)” 

5. In the "Animal Surgery and LIMB" subsection, it is crucial to specify the mouse strain used and detail the rearing conditions.

We have added additional details to the methods to enhance the clarity of the mouse model (pages 5-6, lines 108-122):

“All animal research was performed under protocols approved by the University of Rochester Committee on Animal Resources (UCAR-2019-015). We obtained Catchup mice (C57BL/6 genetic background) (6) from Dr. Minsoo Kim (University of Rochester Medical Center) and maintained the colony. For this study, we used both male and female mice that were 12-20 weeks of age. Thirty minutes before surgery, the mice were given Buprenorphine SR (1 mg/kg) subcutaneously. The mice were then anesthetized with xylazine (10 mg/kg) and ketamine (100 mg/kg) administered intraperitoneally. The surgical procedure and the imaging setup were based on protocols previously described.(4) Briefly, the right femur was implanted with a customized LIMB system. The implant enabled imaging of a fixed region of interest (ROI) proximal to an L-shaped transcortical pin in the diaphyseal bone marrow. The mice received either a sterile pin or a USA300 contaminated pin. 

The mice were imaged at 2-, 4-, and 6-hours post-implantation while anesthetized. During imaging, the mice were lying prone, and the LIMB implant was connected to a custom adaptor for proper placement of the objective lens to find the same ROI. The animals were euthanized by CO2 overdose followed by cervical dislocation to ensure euthanasia.”

6. Regarding Figure 2, does "n=25-57 tracks" introduce potential statistical bias? An elaboration on this aspect would be beneficial.

We thank the Reviewer for raising this question. We would like to clarify that in Figure 2, the differences in the number of tracks (n) are from different mice but not from different users (For mice 1, 2, and 3, User 1 computed 36, 25, and 56 tracks while User 2 computed 40, 25, and 57 tracks respectively). As the data does not follow a normal distribution, we then employed a non-parametric statistical test. In addition, because we observed substantial data spread (high inter-quartile range), which may result in statistical insignificance for small n, we calculated the interclass correlation coefficients (ICC). For a given parameter, ICC takes into account the variability for each individual track giving an accurate estimate of user variability. We calculated ICC to quantify variability in manual tracking and the values ranged from 0.85-0.99 which indicates modest inter-person agreement. We have added ICC for manual tracking to the Results section.

“The major limitation of manual tracking is high inter-user variability. The two users assigned to analyze the same IV-MLSM video often identified different tracks. The lack of consistency is shown in the data distribution, where the median displacement and velocity of neutrophils differed by 25% and 22% respectively between the two users (S2 Fig). While these results were not statistically significant, such substantial numerical differences impede reliable conclusions. Furthermore, the inter-class correlation coefficient for manual tracking was as low as 0.85, indicating only a modest user agreement.”

7. The legend for Figure 2 should include detailed explanations of terms such as "ns" to improve reader comprehension.

We thank the Reviewer for pointing this out. To address this, we added the clarification (“ns = not significant”) to the figure legends.

“Fig 2. The tracking protocol resulted in low inter-person variability in generating tracks. Two users independently analyzed three IV-MLSM timelapse videos and the generated track parameters were compared. Representative 2D IV-MLSM image with overlaid tracks generated by user 1 (A) and user 2 (B). The differences in the tracks generated by both users are shown with white arrows. Semiautomated quantification of neutrophil distance traveled (C), displacement (D), mean speed (E), mean velocity (F), and directionality (G), was performed and the data are presented with the median and interquartile range. (ns = not significant as determined by Mann-Whitney tests adjusted by the Holm-Šídák method (n=25-57 tracks. Representative images from N=3 mice)).”

8. The statistical results in Figure 3B need a more thorough description to enhance clarity and understanding.

We thank the Reviewer for pointing this out. As suggested, we added the statistical details to the figure 3B. 

“Fig 3. Increased neutrophil swarming proximal to MRSA contaminated vs. sterile bone implant. Catchup mice were challenged with a sterile or MRSA-contaminated transfemoral implant and neutrophil swarming behaviors were quantified from 30min IV-MLSM videos obtained at the indicated time post-implantation. (A) Representative 2D IV-MLSM images with overlaid neutrophil tracks proximal to sterile and infected pin at 2-, 4-, and 6-hours post-implantation (S1-3 Videos). (B) Change in neutrophil volume with time proximal to infected and sterile implants calculated using Imaris (S1 Fig). Data is shown as mean ± standard deviation. Unpaired t-tests were used to test the differences between infected and sterile conditions (n=3). Semiautomated quantification of neutrophil distance traveled (C), displacement (D), mean speed (E), mean velocity (F), and directionality (G), was performed and the data are presented with the median and interquartile range. (*p < 0.05 as determined by Mann-Whitney tests adjusted by the Holm-Šídák method (n=105-316 tracks/group/timepoint, N=3 mice/group)).” 

We also clarified in the Results section that we did not observe statistical differences in neutrophil volumes between infected and sterile conditions.

“Consistent with the infection status, the neutrophil volume proximal to the pin increased with time in the infected animals but remained consistent in uninfected animals (Fig 3A-B, S1-3 Videos). However, these results were not statistically significant.”

To improve the understanding of these results in Figure 3, we expanded our discussion. (pages 15-16, lines 332-340)

“Using the LIMB system, we studied neutrophil behavior in response to sterile or MRSA-contaminated implants. As expected, the neutrophil volume proximal to the infected pin was higher compared to that of the sterile pin indicating an active response to infection. Furthermore, in the infected mice, the neutrophils traveled farther (higher distance and displacement) and faster (higher speed and velocity) especially at later time points potentially indicating more potent activation and chemotactic signals compared to the uninfected mice. Future studies with a focus on the quantification of these signals will help elucidate the mechanisms of neutrophil activation and migration. Then the tracking metrics can be validated against these biological outcomes.” 

9. On line 279, "error-prone regions (e.g., cells with small displacement, low intensity, or located at the boundaries of the field of view)" are mentioned. Given these limitations, it is crucial to discuss how the reliability of the results is assured through user verification and the protocol's capabilities.

The Reviewer raised a valid concern. To address this, we included additional descriptions in the Discussion (please refer to the response to comment 2). Specifically, these error-prone regions are corrected manually by the user (Figure 1A, 1E, 1F). To verify if correcting these incorrect tracks results in user bias, we tested the reproducibility of our protocol. As reported in Figure 2, despite these error-prone regions, our protocol remains highly reproducible. Furthermore, we added an add

---

## [Decision Letter · Decision Letter 1]

7 May 2024

A semi-automated cell tracking protocol for quantitative analyses of neutrophil swarming to sterile and S. aureus contaminated bone implants in a mouse femur model

PONE-D-23-40619R1

Dear Dr. Yeh,

We’re pleased to inform you that your manuscript has been judged scientifically suitable for publication and will be formally accepted for publication once it meets all outstanding technical requirements.

Kind regards,

Shigao Huang

Academic Editor

PLOS ONE

Additional Editor Comments (optional):

Reviewers' comments:

Reviewer's Responses to Questions

**Comments to the Author**

1. If the authors have adequately addressed your comments raised in a previous round of review and you feel that this manuscript is now acceptable for publication, you may indicate that here to bypass the “Comments to the Author” section, enter your conflict of interest statement in the “Confidential to Editor” section, and submit your "Accept" recommendation.

Reviewer #1: All comments have been addressed

Reviewer #2: All comments have been addressed

2. Is the manuscript technically sound, and do the data support the conclusions?

Reviewer #1: Yes

Reviewer #2: Yes

3. Has the statistical analysis been performed appropriately and rigorously? 

Reviewer #1: Yes

Reviewer #2: Yes

4. Have the authors made all data underlying the findings in their manuscript fully available?

Reviewer #1: Yes

Reviewer #2: Yes

5. Is the manuscript presented in an intelligible fashion and written in standard English?

Reviewer #1: Yes

Reviewer #2: Yes

6. Review Comments to the Author

Reviewer #1: All of my comments have been well addressed by the authors, I suggest its acceptance for publication.

Reviewer #2: (No Response)

7. PLOS authors have the option to publish the peer review history of their article (what does this mean?). If published, this will include your full peer review and any attached files.

Reviewer #1: **Yes: **yicheng zhao

Reviewer #2: No

---

## [Editor Report · Acceptance letter]

16 May 2024

PONE-D-23-40619R1 

PLOS ONE

Dear Dr. Yeh, 

I'm pleased to inform you that your manuscript has been deemed suitable for publication in PLOS ONE. Congratulations! Your manuscript is now being handed over to our production team.

Kind regards, 

on behalf of

Dr. Shigao Huang 

Academic Editor

PLOS ONE